# Fault Diagnosis of Rotating Machinery Using Kernel Neighborhood Preserving Embedding and a Modified Sparse Bayesian Classification Model

**DOI:** 10.3390/e25111549

**Published:** 2023-11-16

**Authors:** Lixin Lu, Weihao Wang, Dongdong Kong, Junjiang Zhu, Dongxing Chen

**Affiliations:** 1School of Mechatronic Engineering and Automation, Shanghai University, 99 Shangda Road, Baoshan District, Shanghai 200444, China; lulixin@shu.edu.cn (L.L.); weihaowang2020@163.com (W.W.); chendongxing@shu.edu.cn (D.C.); 2College of Mechanical and Electrical Engineering, China Jiliang University, Hangzhou 310018, China; zjj602@yeah.net

**Keywords:** fault diagnosis, rotating machinery, dimension-increment technique, kernel neighborhood preserving embedding, sparse Bayesian classification

## Abstract

Fault diagnosis of rotating machinery plays an important role in modern industrial machines. In this paper, a modified sparse Bayesian classification model (i.e., Standard_SBC) is utilized to construct the fault diagnosis system of rotating machinery. The features are extracted and adopted as the input of the SBC-based fault diagnosis system, and the kernel neighborhood preserving embedding (KNPE) is proposed to fuse the features. The effectiveness of the fault diagnosis system of rotating machinery based on KNPE and Standard_SBC is validated by utilizing two case studies: rolling bearing fault diagnosis and rotating shaft fault diagnosis. Experimental results show that base on the proposed KNPE, the feature fusion method shows superior performance. The accuracy of case1 and case2 is improved from 93.96% to 99.92% and 98.67% to 99.64%, respectively. To further prove the superiority of the KNPE feature fusion method, the kernel principal component analysis (KPCA) and relevance vector machine (RVM) are utilized, respectively. This study lays the foundation for the feature fusion and fault diagnosis of rotating machinery.

## 1. Introduction

Rotating machinery plays an important role in the development of modern industry. The faults will cause shutdown. The timely and effective fault diagnosis of rotating machinery is good for reducing maintenance costs and shorting the shutdown time. Thus, it is important to develop effective fault diagnosis systems for rotating machinery, and artificial intelligence (AI) [1] has been widely utilized for the fault diagnosis of rotating machinery, such as bearings, gears (or gearboxes), engines and turbines. There are two kinds of AI methods for fault diagnosis: machine learning [2] and deep learning [3], as shown in Figure 1. As for machine learning, it consists of feature engineering and shallow learning. It should be noted that feature engineering generally needs experience to select ‘good’ features, and consider the time-consuming of modeling, thus, the feature engineering of machine learning is the research object in this work.

Feature engineering plays a significant role for the predictive performance of the fault diagnosis system. From the previous studies, it can be found that feature engineering mainly includes three aspects: data preprocessing, feature extraction, and feature transformation. As for feature extraction, the commonly used are the time-domain, the frequency-domain, and the time-frequency domain features [4]. It should be noted that the commonly used signal decomposition techniques can provide numerous time-frequency domain features for the fault diagnosis of rotating machinery, such as wavelet transform (WT) [5], wavelet package transform (WPT) [6,7,8], and empirical model decomposition (EMD) [6,9,10,11,12,13]. Normally, the time-frequency domain features are obtained by extracting statistical features, energy, energy ratio, energy entropy or Shannon entropy from sub-signals that are obtained by WT, WPT or EMD. Moreover, excellent feature extraction methods also need to be developed for further improving the diagnostic performance of the decision-making system. Muruganatham et al. [14] applied a singular spectrum analysis (SSA) to extract fault features of roller element bearing. Singular values (SV) and energy of the corresponding principal components of the selected SV were adopted as fault features, respectively. Experimental results showed that the presented method was simple, noise immune and efficient. Liu et al. [15] proposed a short-time matching method based on a kind of atomic decomposition (i.e., matching pursuit). Experimental results showed that the proposed method outperformed the traditional time-domain methods in detecting a bearing incipient fault. Cai et al. [16] proposed a sparsity-enabled signal decomposition method by the combination of a tunable Q-factor WT and morphological component analysis (MCA). Experimental results showed that the proposed method outperformed EMD and spectral kurtosis in identifying a gearbox fault. Li et al. [17] proposed a novel extraction method of deep representation features by utilizing a Boltzmann machine to fuse statistical parameters of WPT. Experimental results showed that the proposed method outperformed the corresponding shallow representations in gearbox fault diagnosis. Gao et al. [18] applied an empirical wavelet transformation (EWT) in time series forecasting. Different from discrete WT, EWT analyzes the data in Fourier domain and implements the spectrum separation, which may make it suitable for analyzing complex and non-stationary vibration signals, especially for rotating machinery. From the previous studies [14,15,16,17,18], it can be found that how to obtain effective fault features through feature engineering or representation learning is an extremely important and complex task in the fault diagnosis of rotating machinery.

In general, the extracted features contain some invalid and redundant features, which will affect the efficiency and accuracy of modeling. Thus, feature transformation is necessary to remove the noise and redundancy features, to improve the predictive performance of rotating machinery fault diagnosis system. Feature selection can remove invalid features that are not sensitive to fault information of rotating machinery. The commonly used feature selection methods include classification accuracy evaluation [11], distance evaluation technique [12] and genetic algorithm (GA) [19,20]. Compared with feature selection, feature fusion is a better method, which is conductive to removing noise and redundancy simultaneously. The commonly used feature fusion methods include principal components analysis (PCA) [7], kernel PCA (KPCA) [7], and manifold learning [5,6,7,13]. Yu [5] utilized the locality preserving projection (LPP) to fuse the original features for the bearing fault diagnosis and found that LPP performed better than PCA. Li et al. [6] utilized locally linear embedding (LLE) to extract distinct features for gear fault diagnosis and found that LLE performed a little better than PCA (less than 1%). In the identification of different gear crack levels, Wan et al. [7] utilized five methods (PCA, KPCA, ISOMAP, LLE and Laplacian Eigenmaps) to carry out dimension-reduction and found that PCA performed the best. Moreover, they pointed out that more case studies should be investigated so as to check whether PCA was still effective. Tang et al. [13] applied orthogonal neighborhood preserving embedding (NPE) for dimension-reduction with regard to turbine fault diagnosis, and found that orthogonal NPE (ONPE) performed better than LPP and LLE. It is worth noting that PCA and KPCA belong to global-preserving techniques while manifold learning (LPP, LLE, NPE, ISOMAP, and Laplacian Eigenmaps) belong to local-preserving techniques [5,21]. In addition to [5,6], image recognition tasks also proved that local-preserving techniques were more suitable for classification problems [22,23]. Thus, manifold learning is recommended for dimension-reduction in the fault diagnosis of rotating machinery. However, how to determine the model parameters and the dimension of the fused features of manifold learning are two thorny problems.

As for the construction of a decision-making system, many machine learning methods have been utilized for fault diagnosis of rotating machinery, such as *k*-nearest neighbors (*k*-NN) [6,11], artificial neural network (ANN) [8,14,19,20], support vector machine (SVM) [9,12,13,15], least square SVM (LS-SVM) [10], and the hidden Markov model (HMM) [24]. In these methods, *k*-NN is easy to understand and implement. However, *k*-NN requires a large amount of calculation since the distance from test points to all samples must be calculated to obtain the *k* nearest neighbors. Moreover, *k*-NN is very vulnerable to sample unbalance. Objectively speaking, ANN, SVM and HMM are more pertinent choices. Nevertheless, how to determine the hyper parameters of ANN, SVM and HMM is a thorny problem, especially for novice researchers. Moreover, these methods [6,8,9,10,11,12,13,14,15,19,20,24] do not belong to a true sparse model, since all features of test points need to participate in calculation. Thus, on the premise of guaranteeing the diagnostic performance, how to develop a true sparse model and greatly reduce the test time is a great challenge.

In this work, we intend to study from two aspects: feature fusion and model construction, about fault diagnosis of rotating machinery. As for the traditional manifold learning methods [5,6,7,13], the model parameters and the dimension of the fused features are hard to determine. To solve these problems, a novel dimension-increment technique is proposed for feature fusion. Kernel neighborhood preserving embedding (KNPE) is realized by the combination of KPCA [25] and NPE [26]. KNPE is conductive to enriching the valid information related to a rotating machinery fault by utilizing the dimension-increment method. Moreover, a novel sparse Bayesian classification (Standard_SBC) method is proposed for model construction, which aims to optimize hyper parameters, overcome non-sparsity and reduce time consumption. Standard_SBC is a variant version of the relevance vector machine (RVM) [27], by removing the kernelization operation. This gives Standard_SBC two advantages: (1) no kernel parameter needs to be optimized in model construction; (2) it automatically selects out more important features in model construction, i.e., integrating feature selection into model construction. This avoids the process of optimization of kernel parameter and feature selection, which greatly reduces the time-consumption of system modeling and helps to realize rapid modeling. In the previous studies [6,7,8,9,10,11,12,13,14,15,19,20,24], determination of the features is independent of the model construction for the fault diagnosis of rotating machinery. It is worth noting that the combination of KNPE and Standard_SBC avoids the determination of feature dimension (a problem that must be solved in traditional dimension-reduction methods), which greatly reduces the workload of feature engineering. In short, the fused features of KNPE are adopted as the input of Standard_SBC, with the aim to construct a more effective SBC-based fault diagnosis system. To show the superiority of KNPE and Standard_SBC, KPCA and RVM are utilized for feature fusion and model construction, respectively.

In this work, two application cases are analyzed to show the effectiveness of the presented method (KNPE + Standard_SBC). The first one is rolling bearing fault diagnosis. The experimental data are obtained from the Bearing Data Center of Paderborn University [28,29]. Another one is the rotating shaft fault diagnosis. The manned recreational facility “Pirate Ship” is popular because of its extreme overweight and weightlessness experience. How to ensure the safe operation of the pirate ship is the top priority. The pirate ship is a kind of large semi-rotating machinery, and the rotating shaft of the suspended hull is subjected to tension and friction for a long time. Once the rotating shaft of the suspended hull fails, it will cause equipment damage and even casualties. Thus, the rotating shaft fault diagnosis is necessary to ensure the safe operation of the pirate ship.

The following are the main contributions of our paper:(1)A new classification method abbreviated as Standard_SBC is firstly proposed, which aims at rapidly constructing the sparse diagnostic model.(2)A new dimension-increment technique abbreviated as KNPE is firstly proposed, which aims at enriching the valid information related to the rotating machinery fault.(3)Standard_SBC can automatically select out more important features from the fused features of KNPE, which greatly simplifies the SBC-based model. The combination of KNPE and Standard_SBC is conductive to rapidly constructing an effective and feasible fault diagnosis system for rotating machinery.(4)A superior diagnostic performance is demonstrated by Standard_SBC when supported by KNPE. Two case studies (on the rolling bearing fault diagnosis and rotating shaft fault diagnosis) validate the effectiveness of KNPE and Standard_SBLR fault diagnosis systems.

The following is the organization of this paper. The background of KNPE and SBC is presented in Section 2. Section 3 presents the fault diagnosis system based on KNPE and Standard_SBC. Section 4 analyzes the presented model and presents the experimental results. Other details about KNPE, Standard_SBC, and future research are provided in Section 5. The paper is concluded in Section 6.

## 2. Backgrounds

The purpose of this section is to provide an introduction to KNPE and Standard_SBC, which will be utilized in Section 4 for the diagnosis of rotating machinery faults.

### 2.1. Neighborhood Preserving Embedding

In order to maintain the local manifold structure of the given data, neighborhood preserving embedding (NPE) [26] is a dimension-reduction technique. In the same manner as locality preserving projection (LPP) [22], NPE is also an approximation to the nonlinear locally linear embedding (LLE) [30]. There are significant differences between NPE and LPP in the objective function.

Given a dataset $X={\left\{x_{i}\right\}}_{i=1}^{m}\in R^{n}$, NPE seeks to find a transformation matrix A that maps these data points $x_{i}$ to a dataset ${\left\{y_{i}\right\}}_{i=1}^{m}\in R^{d}\left(d\ll n\right)$, where $y_{i}=x_{i}A$ is a row vector. The implementation of NPE follows the following steps:

**Step 1:** Constructing the adjacency graph: Assume that G is an adjacency graph with m nodes (i.e., data points). A graph adjacency G is constructed to model the local structure using k-nearest neighbors. Nodes i and j are connected, if $x_{i}$ is one of $x_{j}$’s k-nearest neighbors.

**Step 2:** Calculating the weights: In W, the weights of the edges connecting nodes i and j are represented by $W_{ij}$. The weights of the edges [30] can be obtained by solving Equation (1), which is a sparse but asymmetric matrix of weights.
(1)$$min\sum_{i}{\left\|x_{i}-\sum_{j}W_{ij}x_{i}\right\|}^{2}$$
where $\sum_{j}W_{ij}=1,j=1,2,\cdots ,m$.

**Step 3:** Eigenmaps: Utilizing the Equation (2), the generalized eigenvector problem can be solved to obtain eigenvectors and eigenvalues.
(2)$$X^{T}MX\alpha =\lambda X^{T}X\alpha$$
(3)$$M={\left(I-W\right)}^{T}\left(I-W\right)$$
where $I=diag\left(1,\cdots ,1\right)$, M represents the sparse symmetric and semi-positive definite matrix.

The obtained eigenvectors $\alpha_{1},\;\alpha_{2},\cdots ,\alpha_{d}$ are arrayed according to the ascending order of the obtained eigenvalues $\lambda_{1}\leq \lambda_{2}\leq \cdots \leq \lambda_{d}$. $A=\left(\alpha_{1},\;\alpha_{2},\cdots ,\alpha_{d}\right)$ is a n×d matrix.

Utilizing the NPE, the fused features of the vector $x_{i}$ or the new test data $x_{t}$ are as follows:(4)$$y_{i}=x_{i}A$$
(5)$$y_{t}=x_{t}A$$
where $y_{i}$ and $y_{t}$ are d-dimensional vectors $\left(d\ll n\right)$.

### 2.2. Kernel Neighborhood Preserving Embedding

Kernel neighborhood preserving embedding (KNPE) is the nonlinear form of NPE, which is realized by the combination of KPCA [25] and NPE [26]. Different from the traditional dimension-reduction methods, KNPE is a new dimension-increment technique.

Suppose $\Phi :\;x_{i}\to \Phi \left(x_{i}\right)$ represents the nonlinear mapping from a Euclidean space into a Hilbert space. $X={\left\{x_{i}\right\}}_{i=1}^{m}\in R^{n}$ represents the before-mapping data matrix. $\Phi \left(X\right)={\left[\Phi \left(x_{1}\right),\;\Phi \left(x_{2}\right),\;\cdots ,\;\Phi \left(x_{m}\right)\right]}^{T}\in R^{m\times \infty }$ represents the after-mapping data matrix. Suppose that the data samples ${\left\{\Phi \left(x_{i}\right)\right\}}_{i=1}^{m}$ mapped into the Hilbert space have been centralized, i.e., $\sum_{i=1}^{m}\Phi \left(x_{i}\right)=0$. It is possible to express the generalized eigenvector problem in the Hilbert space in the method as Equation (2):(6)$$\Phi^{T}\left(X\right)M\Phi \left(X\right)\nu =\lambda \Phi^{T}\left(X\right)\Phi \left(X\right)\nu$$
where λ represents the eigenvalue; $\nu \in R^{\infty }$ represents the eigenvector.

The trouble is that the explicit solution to Equation (6) is not available since the nonlinear mapping Φ is implicit and unknown. However, the eigenvector ν can be linearly expressed by Equation (7).
(7)$$\nu =\sum_{i=1}^{m}\alpha^{i}\Phi^{T}\left(x_{i}\right)=\Phi^{T}\left(X\right)\alpha$$
where, $\alpha ={\left[\alpha^{1},\alpha^{2},\cdots ,\alpha^{m}\right]}^{T}$.

Fortunately, the eigenvector α can be obtained by utilizing the kernel tricks function. Equation (8) can be obtained by fusing Equations (6) and (7).
(8)$$\Phi \left(X\right)\Phi^{T}\left(X\right)M\Phi \left(X\right)\Phi^{T}\left(X\right)\alpha =\lambda \Phi \left(X\right)\Phi^{T}\left(X\right)\Phi \left(X\right)\Phi^{T}\left(X\right)\alpha$$

Equation (8) can be simplified and re-expressed, as given by:(9)$$K_{center}MK_{center}\alpha =\lambda K_{center}K_{center}\alpha$$
where $K_{center}=\Phi \left(X\right)\Phi^{T}\left(X\right)$ represents the centralized kernel matrix, λ represents the eigenvalue, and $\alpha \in R^{m}$ represents the eigenvector.

Construction of the centralized kernel matrix $K_{center}$ is the same as that in KPCA [31].
(10)$$K_{center}=K-1_{m\times m}K-K1_{m\times m}+1_{m\times m}K1_{m\times m}$$
(11)$$K={\left[K_{ij}\right]}_{m\times m}$$
(12)$$K_{ij}=K\left(x_{i},x_{j}\right)=\langle \Phi \left(x_{i}\right),\;\Phi \left(x_{j}\right)\rangle$$
where K represents the kernel matrix, $1_{m\times m}$ represents a m×m matrix with the elements 1/m, and $K\left(x_{i},\;x_{j}\right)$ represents the kernel function.

The solution for Equation (9) is available since the centralized kernel matrix $K_{center}$ can be derived easily. Suppose $\left(\alpha_{1},\alpha_{2},\cdots ,\alpha_{m}\right)$ represent the eigenvectors of Equation (9). $V_{d}=\left(v_{1},\;v_{2},\cdots ,\;v_{d}\right)$ represent the first d eigenvectors of Equation (6), which correspond to the first d eigenvalues of Equation (9).

The fused features of $x_{i}$ (i.e., training data) by utilizing KNPE are given by:(13)$$y_{i}=\Phi \left(x_{i}\right)V_{d}=K_{center}^{i}\left(\alpha_{1},\alpha_{2},\cdots ,\alpha_{d}\right)$$
where $K_{center}^{i}$ represents the ith vector of $K_{center}$. d represents the dimension of the matrix.

As for the $x_{t}$, the new test points constructed a new centralized kernel matrix $K_{test}^{C}$ by means of $K\left(x_{t},x_{i}\right)$ and $K\left(x_{i},x_{j}\right)$, which is the same as that in KPCA [31].
(14)$$K_{test}^{C}=K_{test}-K_{test}1_{m\times m}-1_{t\times m}K+1_{t\times m}K1_{m\times m}$$
where $1_{t\times m}$ represents a t×m matrix with the elements 1/m.

The fused features of the new test points $x_{t}$ by utilizing KNPE are given by:(15)$$y_{t}=\Phi \left(x_{t}\right)V_{d}=K_{test}^{C}\left(\alpha_{1},\alpha_{2},\cdots ,\alpha_{d}\right)$$

The dimension of the KNPE-based features can be determined by Equation (16):(16)$$\frac{\sum_{i=1}^{d}\lambda_{i}}{\sum_{i=1}^{m}\lambda_{i}}\geq \gamma$$
where 0<γ≤100%.

The kernel function of KNPE is set as the radial basis function (RBF),
(17)$$K_{RBF}\left(x,y\right)=exp\left(-\frac{{\left\|x-y\right\|}^{2}}{2\sigma_{0}^{2}}\right)$$
where $\sigma_{0}$ represents the kernel parameter, $\sigma_{0}>0$.

### 2.3. Sparse Bayesian Classification

Sparse Bayesian classification (SBC) [27] aims to solve the problem of binary classification, which follows an essentially identical framework as a sparse Bayesian regression (SBR) [27,31]. For the new test points, SBC can provide not only the category output but also the probability. The SBC is introduced as follows.

#### 2.3.1. Probabilistic Model for Classification

Provided a dataset ${\left\{x_{i},\;t_{i}\right\}}_{i=1}^{N}$, $x_{i}\in R^{d}$ is the input variable, $t_{i}\in R$ is the corresponding target value, and N represents the number of data samples. For binary classification, $t_{i}\in \left\{0,1\right\}$ is specified for each sample.

For the new test point $x_{s}$, the posterior probability of $t_{s}=1$ can be expressed by Equation (18).
(18)$$P\left(t_{s}=1|w\right)=\sigma \left\{y\left(x_{s};w\right)\right\}=\frac{1}{1+e^{-y\left(x_{s};w\right)}}$$
(19)$$y\left(x_{s};w\right)=\sum_{i=1}^{N}w_{i}K\left(x_{s},x_{i}\right)+w_{0}$$
where $y\left(x_{s};w\right)$ represents the discriminant function, $w_{i}$ represents the ith element of the weight vector w, and $K\left(x_{s},\;x_{i}\right)$ represents the kernel function.

The category information of the new test point $x_{s}$ is: (1) $t_{s}=0$ when $P\left(t_{s}=1|w\right)<0.5$ is satisfied; (2) $t_{s}=1$ when $P\left(t_{s}=1|w\right)>0.5$ is satisfied.

The discriminant function as given by Equation (19) can be simplified and expressed by:(20)y=Φw
where $y={\left(y_{1},y_{2},\cdots ,y_{N}\right)}^{T}$, $y_{i}=y\left(x_{i};w\right)$, and $\Phi ={\left[\phi \left(x_{1}\right),\phi \left(x_{2}\right),\cdots ,\phi \left(x_{N}\right)\right]}^{T}\in R^{N\times \left(M+1\right)}$ represents the ‘design’ matrix; $\phi \left(x_{i}\right)$ represents the ‘basis function’, $\phi \left(x_{i}\right)=\left[K\left(x_{i},x_{1}\right),K\left(x_{i},x_{2}\right),\cdots ,K\left(x_{i},x_{N}\right)\right]$, and $w={\left(w_{0},w_{1},\cdots ,w_{M}\right)}^{T}$, M=N.

#### 2.3.2. The Fundamentals of SBC

In SBC, the weight coefficients (see Equation (19)) of the binary classification model (see Equation (18)) are considered to be random variables. SBC aims to obtain the posterior distribution over the weights w by means of Laplace approximation so as to construct a binary classification model. A detailed solution process for the weights $w={\left(w_{0},w_{1},\cdots ,w_{N}\right)}^{T}$ is introduced as follows.

$P\left(t|x\right)$ is supposed to satisfy the Bernoulli distribution. The corresponding probability can be calculated by:(21)$$P\left(t|w\right)=\prod_{i=1}^{N}{\sigma \left\{y\left(x_{i};w\right)\right\}}^{t_{i}}{\left[1-\sigma \left\{y\left(x_{i};w\right)\right\}\right]}^{1-t_{i}}$$
where $t={\left(t_{1},t_{2},\cdots ,t_{N}\right)}^{T}$ is the target vector.

In SBC, the prior distribution over the weights w is the same as in SBR [27,31], which is defined as given by:(22)$$p\left(w|\alpha \right)=\prod_{i=0}^{M}N\left(w_{i}|0,\;\alpha_{i}^{-1}\right)$$
where $\alpha =\left(\alpha_{0},\;\alpha_{1},\;\cdots ,\;\alpha_{M}\right)$ consists of M+1 hyper-parameters; $p\left(w_{i}|\alpha_{i}\right)=N\left(w_{i}|0,\alpha_{i}^{-1}\right)$.

For the value of α, the weights $w_{MP}$ can be found by maximizing the log-probability of $p\left(w|t,\;\alpha \right)\propto P\left(t|w\right)p\left(w|\alpha \right)$.
(23)$$log\left\{P\left(t|w\right)p\left(w|\alpha \right)\right\}=\sum_{i=1}^{N}\left[t_{i}\log z_{i}+(1-t_{i})\log(1-z_{i})\right]-\frac{1}{2}w^{T}Aw$$
where $z_{i}=\sigma \left\{y\left(x_{i};w\right)\right\}$, $A=diag\left(\alpha \right)$ is a diagonal matrix.

When the weights $w_{MP}$ are determined, the probability of the test point $x_{i}^{new}$ can be calculated by Equation (18). The corresponding category is given by Equation (24):(24)$${\left.t_{i}\right|}_{x_{i}^{new}}=\left\{\begin{array}{c} 0,\;\;\;P\left(t_{i}=1|w_{MP}\right)<0.5\\ 1,\;\;\;P\left(t_{i}=1|w_{MP}\right)>0.5 \end{array}\right.$$

#### 2.3.3. Hyper-Parameter Estimation of SBC

It should be noted that the weights w cannot be integrated directly. Thus, Laplace’s method is used to obtain an approximate solution. The process is as follows.
The second-order Newton method is utilized to find the values $w_{MP}$.

(25)$$g=\nabla_{w}log\left\{P\left(t|w\right)p\left(w|\alpha \right)\right\}=\Phi^{T}\left(t-z\right)-Aw$$(26)$$H=\nabla_{w}\nabla_{w}log\left\{P\left(t|w\right)p\left(w|\alpha \right)\right\}=-\left(\Phi^{T}B\Phi +A\right)$$(27)$$\Delta w=-H^{-1}g$$(28)$$w_{MP}^{new}=w_{MP}+\Delta w$$
where $z={\left(z_{1},z_{2},\cdots ,z_{N}\right)}^{T}$. $B=diag\left(\beta \right)$ is a diagonal matrix, $\beta =\left(\beta_{1},\beta_{2},\cdots ,\beta_{N}\right)$, wherein $\beta_{i}=z_{i}\left(1-z_{i}\right)$.


2.In the Laplace method, a Gaussian function that centered at $w_{MP}$ is adopted to approximate $p\left(w|t,\alpha \right)$. By utilizing the Hessian matrix that was obtained in step 1, we can derive the corresponding covariance matrix.
(29)$$\Sigma ={\left(-{\left.H\right|}_{w_{MP}}\right)}^{-1}={\left(\Phi^{T}B\Phi +A\right)}^{-1}$$3.The hyper-parameters α need to be updated in the same way as in SBR [27,31].


(30)$$\alpha_{i}^{new}=\frac{\gamma_{i}}{w_{MPi}^{2}}$$(31)$$\gamma_{i}=1-\alpha_{i}\Sigma_{ii}$$
where $w_{MPi}$ represents the ith element of the weights $w_{MP}$; $\Sigma_{ii}$ represents the ith diagonal element of the covariance matrix Σ.

The ‘most probable’ values of $w_{MP}$ can be finally determined by a multi-iteration with regard to Equations (28) and (30). The log-probability $log\left\{P\left(t|w\right)p\left(w|\alpha \right)\right\}$ will gradually increase, most of the hyper-parameters $\alpha_{i}$ gradually tend to infinity and the corresponding weight $w_{i}$ tends to zero. When the iterative process is complete, the ‘most probable’ values of $w_{MP}$ will be obtained.

#### 2.3.4. The Proposed Standard_SBC

In the original literature [27], SBC is also referred to as a relevance vector machine (RVM). The discriminant function (see Equation (19)) in RVM is a kernelized model, which is similar to the support vector machine (SVM). In this work, a new SBC method, abbreviated as Standard_SBC, is proposed so as to match with dimension-increment (i.e., KNPE).

Standard_SBC is a variant version of RVM and realized by using a standard linear model as the discriminant function, as given by Equation (32).
(32)$$y\left(x_{s};w\right)=\sum_{i=1}^{d}w_{i}x_{s}^{i}+w_{0}$$
where $w_{i}$ represents the ith row of the weight vector w, $x_{s}^{i}$ represents the ith element of the input variable $x_{s}$.

The discriminant function as given by Equation (32) can also be simplified and expressed by:(33)y=Φw
where $y={\left(y_{1},y_{2},\cdots ,y_{N}\right)}^{T}$, $y_{i}=y\left(x_{i};w\right)$, $\Phi ={\left[\phi \left(x_{1}\right),\phi \left(x_{2}\right),\cdots ,\phi \left(x_{N}\right)\right]}^{T}\in R^{N\times \left(M+1\right)}$ represents the ‘design’ matrix, $\phi \left(x_{i}\right)$ represents the ‘basis function’, $\phi \left(x_{i}\right)=[1,x_{i}]$, $w={\left(w_{0},w_{1},\cdots ,w_{M}\right)}^{T}$, and M=d.

#### 2.3.5. Summary of SBC

It should be noted that Equations (20) and (33) have the same expression. The difference is that they have a different ‘basis function’ $\phi \left(x_{i}\right)$. According to the ‘basis function’ $\phi \left(x_{i}\right)$, SBC can be divided into two categories:(1)When $\phi \left(x_{i}\right)=[1,\;x_{i}]$ is adopted as the ‘basis function’, SBC is defined as Standard_SBC.(2)When $\phi \left(x_{i}\right)=\left[1,K\left(x_{i},x_{1}\right),K\left(x_{i},x_{2}\right),\cdots ,K\left(x_{i},x_{N}\right)\right]$ is adopted as the ‘basis function’, SBC is defined as Kernelized_SBC.

Most of the weight coefficients (see Equations (19) and (32)) tend to zero when the hyper-parameters’ estimation of SBC is completed. This reveals the sparseness of SBC. Thus, it is possible to improve the model framework of SBC in accordance with the number of non-zero weights that are given by:(34)$$y\left(x_{s};w\right)=\sum_{x_{s}^{i}\in RD}w_{i}x_{s}^{i}+w_{0}$$
(35)$$y\left(x_{s};w\right)=\sum_{x_{i}\in RV}w_{i}K\left(x_{s},x_{i}\right)+w_{0}$$
where $w_{i}$ represents the ith element of the weight vector w, $K\left(x_{s},\;x_{i}\right)$ represents the kernel function, $x_{t}^{i}$ represents the ith element of the input variable $x_{t}$.

In summary, SBC can be divided into two categories: Standard_SBC and Kernelized_SBC. Kernelized_SBC is also known as RVM, which was discussed in our previous research [32].

## 3. Fault Diagnosis of Rotating Machinery Based on KNPE and Standard_SBC

In this work, Standard_SBC is utilized to construct the SBC-based fault diagnosis model of rotating machinery. Figure 2 illustrates the overall process.

### 3.1. Feature Extraction and Fusion

Firstly, feature extraction from the signals need to be carried out so as to directly reflect the change of the health state of rotating machinery. The widely used signals for the fault diagnosis of rotating machinery contains vibration, acoustic emission (AE) and current. To remove the influence of different operating conditions, as shown in Table 1, the features are determined by our previous work [21]. The extracted features need to be normalized according to Equation (36) before fusion or feeding to Standard_SBC.
(36)$$x^{\prime }=\frac{x-\bar{x}}{\sigma_{x}}$$
where $\bar{x}$ represents the mean value of x; $\sigma_{x}$ represents the standard deviation of x.

Secondly, to remove as much noise and redundancy as possible, KNPE is utilized to fuse the extracted features. Moreover, the model parameters $\left(k\;\mathrm{and}\;\sigma_{0}\right)$ of KNPE have a certain influence on the effectiveness of the fused features. The pseudo-code of KNPE is shown in Table 2.

### 3.2. Model Construction

The extracted features or the KNPE-based fusion features make up the feature vectors, which are adopted as the input of Standard_SBC.

Pseudo-code of the SBC algorithms is shown in Algorithm 1. After the training process, the parameters of Standard_SBC ($w_{MP}$) can be obtained and utilized to construct a fault diagnosis system based on Standard_SBC. There is no kernel parameter that needs to be optimized in Standard_SBC, which will greatly reduce the modeling time. Moreover, Standard_SBC is a standard linear and sparse model, which will greatly reduce the recognition time for test samples.

Note that Standard_SBC is a binary classifier. A voting method [36] is needed to perform the task of multi-classification.
**Algorithm 1:** Standard_SBC and Kernelized_SBCSBC-training1Constructing the ‘design’ matrix: $\Phi ={\left[\phi \left(x_{1}\right),\;\phi \left(x_{2}\right),\;\cdots ,\;\phi \left(x_{N}\right)\right]}^{T}\in R^{N\times \left(M+1\right)}$
2Initialization of $w={\left(w_{0},\;w_{1},\;\cdots ,\;w_{M}\right)}^{T}$, $\alpha =\left(\alpha_{0},\;\alpha_{1},\;\cdots ,\;\alpha_{M}\right)$ and $\beta =\left(\beta_{1},\beta_{2},\cdots ,\beta_{N}\right)$
3$A=diag\left(\alpha \right)$, $B=diag\left(\beta \right)$4**for** i=1:maxIteration % maxIteration is the maximum number of iterations.5  Calculation of the log-probability $log\left\{P\left(t|w\right)p\left(w|\alpha \right)\right\}$ as given by Equation (23)6  Update of the covariance matrix Σ as given by Equation (29)7  Update of the weights $w_{MP}^{new}$ as given by Equation (28)8  Update of the hyper-parameters $\alpha_{i}^{new}$ as given by Equation (30)9  Update of the diagonal matrix A
10  Update of the diagonal matrix B
11  Calculation of the convergence condition: ${tmp}_{err}=max\left\{\left\|\alpha_{i}^{new}-\alpha_{i}\right\|\right\}$
12  **if** ${tmp}_{err}<10^{-5}$13    **break** % Iteration is stopped when the convergence condition is satisfied.14  
**end if**
15**end for** % The ‘most probable’ values $w_{MP}$ are determined.SBC-prediction1Constructing the ‘design’ matrix $\phi \left(x_{*}\right)$
2Calculation of the probability output for test point $x_{i}^{new}$ as given by Equation (18)3Determination of the category label for test point $x_{i}^{new}$ as given by Equation (24)**Note:** $x_{i}\in R^{d}$ is the input variable. As for Standard_SBC, $\phi \left(x_{i}\right)=\left[1,x_{i}\right]$ and M=d. As for Kernelized_SBC, $\phi \left(x_{i}\right)=\left[1,K\left(x_{i},x_{1}\right),K\left(x_{i},x_{2}\right),\cdots ,K\left(x_{i},x_{N}\right)\right]$ and M=N. In step 6, SVD [34,35] is applied for reducing the singularity of the matrix $\left(\Phi^{T}B\Phi +A\right)$. Detailed process is provided in Appendix A.

## 4. Experimental Results and Analysis

To show the effectiveness of the proposed method (KNPE + Standard_SBC) for fault diagnosis of rotating machinery, two case studies are analyzed in this work.

### 4.1. Case Study 1: Rolling Bearing Fault Diagnosis

#### 4.1.1. Experimental Introduction

In this section, the data are obtained from the Bearing Data Center of Paderborn University [28,29]. The experimental setup for the collection of bearing data with the working condition is shown in Figure 3. The test rig consists of five modules: (1) electric motor, (2) torque-measuring shaft, (3) bearing test module, (4) flywheel, and (5) load motor. The bearing test module is used to generate the experimental data of ball bearings with different types of damage: non-damage (healthy), artificial damages and real damages. The type of electric motor and load motor is Permanent Magnet Synchronous Motor (PMSM). The phase currents $\left(u,v\right)$ of the electric motor are collected by a current transducer, low-pass filter at 25 kHz and A/D converter [29]. The vibrations of the bearing test module are collected by a piezoelectric accelerometer, charge amplifier, low-pass filter at 30 kHz, and A/D converter [29]. The sampling frequency for the phase currents $\left(u,v\right)$ and vibration information are set to 64 kHz. Further details of the experimental setup are available in [29].

Real damages caused by accelerated lifetime tests are as shown in Figure 4. The experimental data obtained from 15 bearings are listed in Table 3. For each bearing, the data are collected at different operating parameters, as shown in Table 4. For each kind of setting, a total of 20 measurements are saved as MATLAB files. The name for the MATLAB files consists of the operating parameters, the bearing code and the code of measurement (e.g., N15_M07_F10_K001_1.mat). There are a total of 1200 measurements that can be obtained: 15 bearings × 4 settings × 20 measurements.

Table 5 shows the details of the operating parameter of healthy (undamaged) bearings during the run-in period. And, the Table 6 shows the detail information of bearings with real damages caused by accelerated lifetime test. The extent of the damage describes the size of the damage in normalized levels, which are independent of the bearing size. The levels are based on the length of the damage and the parameters for describing the geometry of bearing damages, as shown in Figure 5. The damage percentage of the length relative to pitch circumference is calculated and then assigned to five levels according to Table 7, especially for bearing 6203.

#### 4.1.2. Feature Extraction and Fusion

Due to the influence of different operating parameters, the original phase currents $\left(u,v\right)$ and vibrations find it difficult to reflect the bearing damages. Thus, the features [21] as shown in Table 1 are extracted. A total of 8 Time−domain×3 Signals=24 time−domain features and 2 Frequency−domain×3 Signals=6 frequency−domain features are obtained for each signal. Moreover, the decomposition of the phase currents $\left(u,v\right)$ and vibrations at level-5 are based on the use of ‘db5′ wavelet packets’ decomposition (WPD) with Shannon entropy [36]. A total of 32 sub-bands can be obtained from the phase currents $\left(u,v\right)$ and vibrations, respectively. By utilizing the WPD, a total of 32 wavelet−domain×3 Signals=96 wavelet−domain features can be obtained by calculating the root mean square (RMS) of the wavelet packet coefficients in each sub-band.

The “feature vector” of samples consists of 24 time-domain (T) features, 6 frequency-domain (F) features and 96 wavelet-domain (W) features. The 126 TFW features or the fused features of KNPE are adopted as the features of the SBC-based rolling bearing fault diagnosis system, as can be observed in Table 8.

The phase currents $\left(u,v\right)$ and vibrations are divided into four segments at the interval of one second, as shown in Figure 6. Segment 1 and segment 3 are utilized to train models. Segment 2 and segment 4 are utilized to test models. Altogether, 4800 data samples can be obtained: 15 bearings × 4 settings × 20 measurements × 4 segments. The samples are divided into a training dataset (2400) and test dataset (2400).

#### 4.1.3. Effectiveness of KNPE

To verify the effectiveness of KNPE, the SBC-based rolling bearing fault diagnosis system is constructed by utilizing the extracted features and the fused features of KNPE, respectively. The detailed process is as follows.

##### Model Construction and Evaluation by Using the Original Features

The Standard_SBC classifier is a binary classifier. This study focuses on multi-classifying bearings with real damages. To perform the task of multi-classification, a one-versus-one method [33], also called voting, is employed. Suppose a binary classifier is trained using two categories of samples. Any two categories of samples can be used to train the classifier. Altogether, $l\left(l-1\right)/2$ binary classifiers can be obtained by the combination of any two categories and utilized to realize an l-classification. Test samples are classified by the $l\left(l-1\right)/2$ binary classifiers. The category with the greatest number of votes from the $l\left(l-1\right)/2$ binary classifiers is considered to be the final estimate.

A pseudo-code of the SBC algorithms is shown in Algorithm 1, and the set of hyper-parameters are shown in Table 9. In step 2 of SBC-training, the weights w are initialized as $w_{0}=w_{1}=\cdots =w_{M}=0$, and the hyper-parameters α are initialized as $\alpha_{0}=\alpha_{1}=\cdots =\alpha_{M}=5\times 10^{-5}$. As for Standard_SBC, the subscript of the weight $w_{M}$ and the hyper-parameter $\alpha_{M}$ refers to the dimension of the “feature vector” $x_{i}\in R^{d}$, i.e., M=d. The details are provided in Section 2.3.4. The parameters β are initialized as $\beta_{1}=\beta_{2}=\cdots =\beta_{N}=0.25,$ since $\beta_{i}=z_{i}\left(1-z_{i}\right)$ and $z_{i}=\sigma \left\{y\left(x_{i};w\right)\right\}$. In the iterative process, most of the hyper-parameters $\alpha_{i}$ gradually tend to infinity while the corresponding weights $w_{i}$ tend to zero. In practical terms, the weights $w_{i}$ have been very close to zero when the corresponding hyper-parameters $\alpha_{i}$ exceed $10^{5}$. Thus, in this work, $\alpha_{i}$ will stop being updated when $\alpha_{i}>10^{5}$ is satisfied. The maximum iterations of Standard_SBC is set to 1000. Moreover, the convergence condition in the iterative process is set to ${tmp}_{err}=max\left\{\left\|\alpha_{i}^{new}-\alpha_{i}\right\|\right\}<10^{-5}$.

Altogether, three Standard_SBC-based binary classifiers are needed to realize a three-classification for the actual state of bearings: model_1-2, model_1-3 and model_2-3. The model_1-2 represents the class1 versus class2 of the Standard_SBC-based classifier.

The iterative process of Standard_SBC can be described by the variation of $log\left\{P\left(t|w\right)p\left(w|\alpha \right)\right\}$, as shown in Figure 7; the horizontal axis represents the number of iterations, and the vertical axis represents the value of $log\left\{P\left(t|w\right)p\left(w|\alpha \right)\right\}$; most of the weights $w_{i}$ for model_1-2, model_1-3, and model_2-3 are equal to zero when the iterative process is finished. Figure 8 shows the weights $w_{i}$ of the SBC-based rolling bearing fault diagnosis system, Figure 8a–c represents the weight of the model_1-2, model_1-3 and model_2-3, respectively. In Figure 8, the horizontal axis represents the index of weights, and the vertical axis represents the value of the weights. Test results are shown in Figure 9. There are only three misclassification samples in 2400 test samples, the horizontal axis represents the index of samples, where index 1–800 represents the samples in state 1, index 801–1600 represents the samples in state 2, index 1601–2400 represents the samples in state 3, the vertical axis represents the state of predicted samples, blue dots represent the correctly classified samples, and red dots represent the misclassified samples. The result shows that the accuracy of the SBC-based rolling bearing fault diagnosis system reaches up to 99.88%.

In the following section, C1 and C2 represent the phase currents u and v of an electric motor, respectively. V represents the vibrations of the bearing test module. Moreover, as mentioned in Table 1, ‘T’ represents the time-domain, ‘F’ represents the frequency-domain, and ‘W’ represents the wavelet-domain.

To comprehensively analyze the performance of the SBC-based rolling bearing fault diagnosis system, different combinations of signals and features (TFW features, TF features, and W features) are analyzed. The number of features extracted by utilizing the different signals and domains are shown in Table 10. The performance evaluation of the SBC-based rolling bearing fault diagnosis system with different combinations is shown in Figure 10, the horizontal axis represents the signals utilized by the models, the vertical axis represents the prediction accuracy, the blue column represents the prediction accuracy by utilizing the TFW features, the red column represents the prediction accuracy by utilizing the W features, and the brown column represents the prediction accuracy by utilizing the TF features; the following conclusions can be drawn from Figure 10:(1)When the features are determined, the combination of vibrations (V) and phase currents (C1+C2, C1 or C2) can guarantee a better diagnostic performance.(2)W features are more important than TF features in eliminating the influence of different operating conditions.

The sparseness of the SBC-based rolling bearing fault diagnosis system by using the TFW features is shown in Figure 11. It can be found that the more signals (C1+C2+V) are utilized to construct models, the less number of ‘relevance dimensions’ (RDs).

In summary, the more sufficient signals (C1+C2+V) and sufficient features (TFW features) are utilized to construct the SBC-based rolling bearing fault diagnosis system, the better diagnostic performance and larger sparseness.

##### Model Construction and Evaluation by Using the Fused Features of KNPE

In this section, to guarantee the diagnostic performance of the SBC-based system, the KNPE is utilized to fuse the features. A pseudo-code of KNPE is shown in Table 2. In the step 1 of KNPE, k-nearest neighbors $\left(k\in N\right)$ is adopted to construct the adjacency graph. In this work, the parameter k and the kernel parameter $\sigma_{0}$ of KNPE are set to 10 and 5, respectively.

KNPE is utilized to fuse the features (TFW features, TF features, and W features) of signals. The first d fused features of KNPE are selected as the input. The dimension d is determined by the cumulative contribution rate γ, as shown in Equation (16). In this section, the parameter γ is set to 50%. The effectiveness of KNPE in improving the diagnostic performance of the SBC-based fault diagnosis system is shown in Figure 12. It can be found that, when the features are sufficient (see Figure 12a) or relatively sufficient (see Figure 12b), KNPE can improve the diagnostic performance of the SBC-based system, especially in the case of less signals (C1+C2, C1, C2 and V). When the features are insufficient (see Figure 12c), vibrations (V) are still effective with the support of KNPE and perform far better than phase currents (C1+C2, C1 or C2). With the combination of vibrations (V) and TF features (see Figure 12c), KNPE makes the prediction accuracy of the SBC-based system improve from 65.88% to 97.33%. This reveals that vibrations (V) are more suitable for rolling bearing fault diagnosis.

In summary, KNPE is conductive to guaranteeing a better diagnostic performance even when the conditions become more severe (i.e., signals or features are reduced). Thus, the combination of KNPE and Standard_SBC is recommended for constructing the SBC-based rolling bearing fault diagnosis system.

### 4.2. Case Study 2: Rotating Shaft Fault Diagnosis

#### 4.2.1. Experimental Introduction

In this section, the rotating shaft of the suspended hull of pirate ship is analyzed. The experimental setup for the collection of rotating shaft data with a working condition is shown in Figure 13. The test rig contains five modules: (1) main frame, (2) suspended hull, (3) bearing pedestal, (4) rotating shaft, and (5) power take-off. The rotating shaft is used to generate the experimental data with different damages: normal shaft (healthy), unbalanced shaft and cracked shaft, as shown in Figure 14. Figure 14a shows the normal shaft, and the maximum outer diameter and maximum length of the shaft are 55 mm and 620 mm, respectively. Figure 14b shows the unbalanced shaft, and the dimensions of the iron blocks are 53 mm in length, 39 mm in width, and 24 mm in height. Figure 14c shows the cracked shaft, and the cracked shaft is artificially formed by cutting a gap with a width of 0.2 mm and length of 20 mm. The vibrations of rotating shafts are collected by an accelerometer (CT1005L), constant current source (CT5201), data acquisition card (MCC USB-231), and computer. The type and function of the key components are shown in Table 11. The accelerometer is utilized to convert vibration signals into voltage signals. The constant current source is utilized to stable the voltage signal or amplify the voltage signal by 10 times. The data acquisition card is utilized to collect the voltage signal. Accelerometer, constant current source, data acquisition card, and computer are wired as shown in Figure 15; the accelerometer and constant current source are wired by a “M5 to BNC” cable. The constant current source and data acquisition card are wired by the “BCN to terminal” cable. The data acquisition card is connected to the computer through a “Hi-speed Micro-USB” cable. The accelerometer is adsorbed on the side of the bearing pedestal through the magnetic base, as shown in Figure 13b.

#### 4.2.2. Data Collection Introduction

In this section, the data collection and the procedure of the experiments have been introduced. The experimental data obtained from the three types of rotating shaft are listed in Table 12. The sampling frequency for the vibration information of the rotating shafts is set to 10 kHz. According to Nyquist’s theorem, we set the cut-off frequency at 4 kHz of a low-pass filter [37]. A total of 330 measurements are obtained and saved as CSV files. In each file, the data length is 30 s. A total of 110 signals were collected for three kinds of the shaft, respectively. A total of 110 signals × 3 shafts = 330 signals were collected. Each signal was divided into 10 segments for the first 5 s. This resulted in a total of 3300 samples, with 1100 samples in each category, and for each category, the 1100 samples have the same level of damage.

The procedure of data collection in the pirate ship experiment as shown in Figure 16; the steps are as follows.

Step 1. Raise the pirate ship to the initial position, with an angle of 45° from the vertical line.

Step 2. Release the pirate ship, and let it swing freely.

Step 3. After the pirate ship swings to the terminal position, let it return freely.

Step 4. The pirate ship returned to the initial position and was blown up with compressed air.

Step 5. Collect the signals of the pirate ship shaft.

Step 6. Repeat the step 3 to step 5.

The initial power for the suspended hull is generated by lifting it to the initial position, with an angle of 45° from the vertical line, and released to swing freely. The follow-up power for the suspended hull is generated by an air valve which will provide the power take-off when the pirate ship swings back to its initial position. The energy was added by blowing air to compensate for the energy lost during the swing process. The accelerometer collects the vibration signals generated by the rotating shaft.

#### 4.2.3. Feature Extraction and Fusion

When shaft faults occur, non-stationary signals and noise signals will appear abundantly. The key of the rotating shaft fault diagnosis is to how to extract effective features and remove the noises simultaneously. In this section, the feature extraction is the same as that in Section 4.1.2. The 8 time-domain (T) features, 2 frequency-domain (F) features and 32 wavelet-domain (W) features, as can be observed in Table 1, are extracted from the measurements (i.e., CSV files). The first 5 s of the data files are adopted for feature extraction. The sampling interval for feature extraction is set to 0.5 s, and can augment 1 sample to 10 samples. Altogether, 3300 samples can be obtained: 10 samples × 110 data files × 3 categories. They are randomly divided into a training dataset (1650) and test dataset (1650). The corresponding shaft fault, as shown in Table 12, is adopted as the target value. The features need to be normalized according to Equation (36).

KNPE is utilized to fuse the features (TFW features, TF features, and W features). The parameter k and the parameter $\sigma_{0}$ of KNPE are set to 10 and 5, respectively. In this section, the cumulative contribution rate γ (see Equation (16)) is set to 2%. KNPE is utilized to extract more effective features and remove the noise and redundancy features.

#### 4.2.4. Effectiveness of KNPE

In this section, the extracted features (TFW features, TF features, and W features) and the fused features of KNPE are adopted for model construction, respectively. Standard_SBC is utilized to realize a rotating shaft fault diagnosis. The effectiveness of KNPE in improving the diagnostic performance of the SBC-based rotating shaft fault diagnosis system is shown in Figure 17. It can be found that KNPE can improve the diagnostic performance of the SBC-based system, especially in the case of less features (TF features). Moreover, W features perform far better than TF features in prediction accuracy. With the case of KNPE fusing the TFW features, the prediction accuracy of the SBC-based rotating shaft fault diagnosis system reaches up to 99.64%. Test results of the rotating shaft fault diagnosis system based on KNPE and Standard_SBC by using the TFW features are shown in Figure 18. There are only six misclassification samples in the 1650 test samples.

### 4.3. Comparison with KPCA

KNPE is a novel feature dimension-increment method. To show the superiority of KNPE, KPCA [25] is utilized to feature fusion. The radial basis function (RBF) (see Equation (17)) is adopted as the kernel function of KPCA. The kernel parameter is also set to $\sigma_{0}=5$. In this section, KPCA is utilized to fuse the TFW features, TF features and W features, respectively. Performance comparison between KNPE and KPCA is carried out when the cumulative contribution rate γ (see Equation (16)) is selected from different orders of magnitude ($10^{-1}\sim 10^{0}$, $10^{0}\sim 10^{1}$, and $10^{1}\sim 10^{2}$). The KNPE-based fusion features and KPCA-based fusion features are adopted as the input, respectively. The same training and test dataset in Section 4.1 and Section 4.2 are adopted for model construction, respectively. The performance of KNPE and KPCA are shown in Figure 19, Figure 20 and Figure 21. The olive dashed lines in some subfigures (see Figure 19c, Figure 20a–c and Figure 21b,c) represent the threshold which is obtained from the SBC-based system by utilizing the pre-fusion features.

Note that the effectiveness of KNPE and KPCA is dependent on sufficient signals, sufficient features (TFW features), or relatively sufficient features (W features). It can be found that, when the features are sufficient (see Figure 19a and Figure 21a) or relatively sufficient (see Figure 19b and Figure 21b), the effectiveness of KNPE is superior to KPCA.

When the signals are sufficient (C1+C2+V), feature absence (see Figure 19c) has little effect on the performance of KNPE. When the signals are reduced (see Figure 20, only the vibrations (V) are adopted), KNPE is still effective with most of the cumulative contribution rates and performs better than KPCA. There are some discontinuities (drops) in the prediction accuracy that corresponds to KNPE, as shown in Figure 20b,c. This may be because Standard_SBC does not find the correct ‘relevance dimensions’ (RDs). On the whole, the stability of the KNPE-based fusion features is less affected by insufficient signal or feature absence, in comparison with KPCA.

It can be found from the prediction accuracy of Figure 19, Figure 20 and Figure 21 that KNPE is effective at most cumulative contribution rates. Furthermore, KNPE is less affected by an insufficient signal or feature absence in comparison to KPCA.

### 4.4. Comparison with RVM

To show the superiority of Standard_SBC, RVM (i.e., Kernelized_SBC) [32] is utilized to construct the SBC-based system in this section. In RVM, the initialization for the weights w, the hyper-parameters α and the parameters β are the same as that in Section Model Construction and Evaluation by Using the Original Features. Moreover, the maximum iterations and the convergence condition in the iterative process are also the same as that in Section Model Construction and Evaluation by Using the Original Features. The RBF kernel function in RVM is given by:(37)$$K\left(x_{i},\;x_{j}\right)=exp\left(-\varphi \cdot {\left\|x_{i}-x_{j}\right\|}^{2}\right)$$
where φ is the kernel parameter, φ>0.

In RVM, a 3-fold cross-validation and grid-search are utilized to optimize the kernel parameter φ. The interval range of φ is set to $\left[10^{-10},10^{0}\right]$. The Δφ is set to $10^{0.5}$. In the 3-fold cross-validation, the classification rate of the SBC-based system is adopted as an evaluation indicator. An average of the 3 classification rates is adopted as the fitness. The optimal $\varphi^{*}$ that corresponds to the maximum fitness is selected from the one-dimension grid space (formed by φ) when the grid-search is finished.

The same training dataset and test dataset in Section 4.1 and Section 4.2 are adopted for the model construction. As for the rolling bearing fault diagnosis, TFW features of C1+C2+V are adopted. As for the rotating shaft fault diagnosis, TFW features of vibrations are adopted. Moreover, KNPE is utilized for feature fusion so as to further analyze the performance of RVM. The parameter k, the kernel parameter $\sigma_{0}$ (see Equation (17)) and the cumulative contribution rate γ (see Equation (16)) of KNPE are set to 10, 5 and 0.2%, respectively.

Comparative analysis for the time consumption of Standard_SBC and RVM is carried out. The comparison results of Standard_SBC and RVM are listed in Table 13. CV_time represents the time consumption of the parameter optimization for determining the kernel parameter φ of RVM. The training_time represents the amount of time it takes to construct a model by utilizing the determined model parameters. The test_time represents the amount of time it takes to test the model.

The obvious advantage is that Standard_SBC is not required to carry out the kernel parameter optimization in model construction. In addition, with the assumption of guaranteed prediction accuracy, the Test_time of Standard_SBC is significantly less than that of RVM. In summary, Standard_SBC has the advantages of prediction accuracy, less Training_time, and less Test_time, which make it suitable for industrial applications. This explains why Standard_SBC is utilized for model construction in this work.

## 5. Discussion

In this work, Standard_SBC is proposed on the basis of sparse Bayesian classification (SBC). Sparseness is an attribute of SBC. As for Kernelized_SBC (i.e., RVM), sparseness is directed against the training samples. However, as for Standard_SBC, sparseness is directed against the dimensions of training samples. In comparison with Kernelized_SBC, the superiority of Standard_SBC is that no kernel parameter needs to be optimized in model construction, which is conductive to rapid modeling.

It is worth noting that the sparsity connotation of Kernelized_SBC and Standard_SBC is different. As for Kernelized_SBC (i.e., RVM), its sparsity refers to only the ‘relevance vectors’ (RVs) are involved in decision making, as given by Equation (35). However, all the dimensions of the test points need to be involved in the calculation for decision making. As for Standard_SBC, its sparsity refers to only the ‘relevance dimensions’ (RDs) being involved in decision-making, as given by Equation (34), i.e., only a small percentage of the dimensions of the test points need to be involved in the calculation for decision making. From the perspective of the number of the dimensions of the test points involved in decision making, the proposed Standard_SBC belongs to a true sparse predictive model.

Moreover, KNPE is proposed by the combination of KPCA and NPE. In this work, the parameters (k and $\sigma_{0}$) of KNPE are determined by training. There is no doubt that a selection of the model parameters (k and $\sigma_{0}$) will affect the effectiveness of the KNPE-based fusion features. In the following research, the selection and optimization of the parameters (k and $\sigma_{0}$) of KNPE will be carried out.

As mentioned in the previous sections, KNPE is a novel feature dimension-increment method. It means that the dimension of the KNPE-based fusion features may be much larger than that of the original features, which greatly enriches the valid information related to the rotating machinery fault. However, a large number of features will greatly increase the complexity of the diagnostic model. Fortunately, Standard_SBC can automatically select more important features from the fused features of KNPE, which avoids the generation of more complex diagnostic models that may be caused by a dimension-increment operation. This greatly simplifies the SBC-based fault diagnosis system of rotating machinery.

In future research, the fault diagnosis system of rotating machinery based on KNPE and Standard_SBC will be validated through more case studies, such as rotor fault diagnosis and gear fault diagnosis. In addition, KNPE and Standard_SBC will be applied to other fields as well.

## 6. Conclusions and Future Works

### 6.1. Conclusions

In this paper, Standard_SBC is comprehensively analyzed with the aim to construct an effective and feasible SBC-based fault diagnosis system of rotating machinery by drawing support from KNPE. To reveal the effectiveness of the fault diagnosis system of rotating machinery based on KNPE and Standard_SBC, two application cases (rolling bearing fault diagnosis and rotating shaft fault diagnosis) are analyzed in detail. The main conclusions are as follows:(1)Experimental results of rolling bearing fault diagnosis show that the combination of multiple signals is conductive to improving the diagnostic performance of the SBC-based system.(2)Experimental results of the two application cases show that sufficient features (TFW features) are necessary for guaranteeing better diagnostic performance of the SBC-based system.(3)KNPE is conductive to guaranteeing the diagnostic performance of Standard_SBC. When using sufficient features (TFW features) of the vibrations, experimental results of the two application cases show that KNPE can make the prediction accuracy of the SBC-based system more than 99.5%.(4)In comparison with Kernelized_SBC (i.e., RVM), Standard_SBC can realize rapid modeling (no kernel parameter needs to be optimized) and has less testing time, which make it suitable for industrial applications.(5)The cumulative contribution rate of KNPE has a much larger selectable region than KPCA. Therefore, the determination of a suitable cumulative contribution rate for KNPE is much easier than KPCA.

### 6.2. Future Works

It can be found that how to obtain ‘good’ features plays an important role in improving the diagnostic performance of the fault diagnosis system of rotating machinery. For the application, with the efficient advantages of KNPE, the fault of rotating machinery can be found in a timely manner, and the maintenance measures can be taken in a timely manner. The KNPE method can be utilized to monitor the bearing and motor of a high-speed railway, semiconductor production line, generator status, and so on. For future works, KNPE can be combined with ANN, LSTM, and so on to further improve the fault diagnosis accuracy of rotating machinery. Moreover, we can also combine the KNPE with KPCA; firstly, KNPE is utilized to increase the dimension of features, and then KPCA is utilized to decrease the dimension of features; thus, we can extract the most useful features.

A theoretical framework for the diagnosis of faults in rotating machinery for industrial production is provided in this paper.

## Figures and Tables

**Figure 1 entropy-25-01549-f001:**
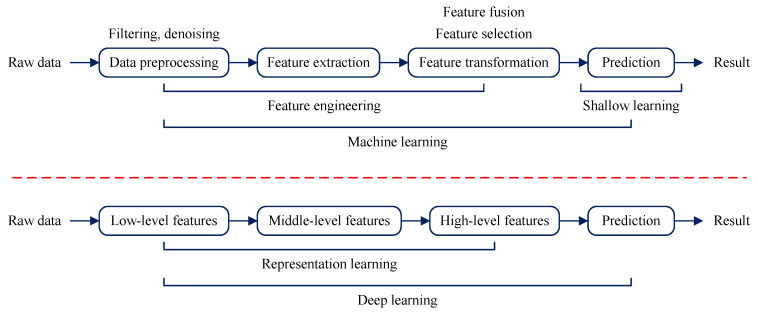
AI methods for fault diagnosis of rotating machinery: (1) Machine learning; (2) deep learning.

**Figure 2 entropy-25-01549-f002:**
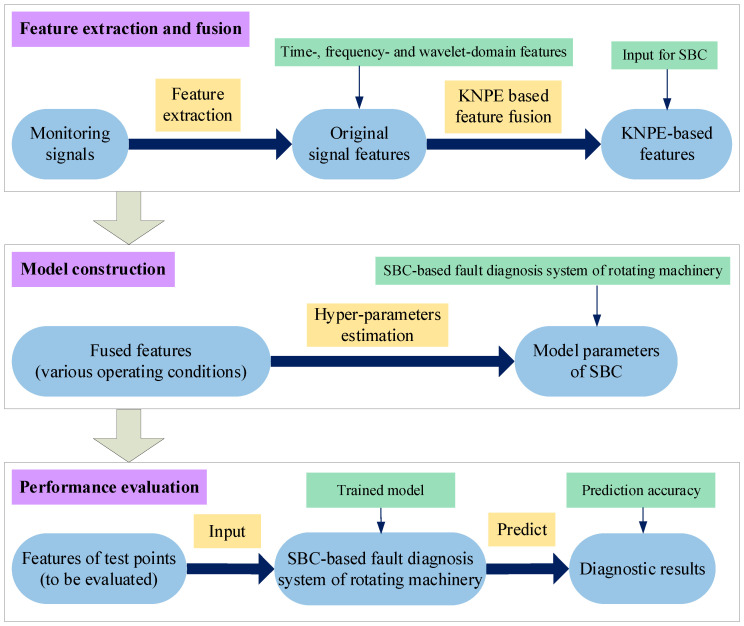
Fault diagnosis system of rotating machinery based on KPNE and Standard_SBC. The small two-way arrows mean that both sides are equivalent to each other. The small single arrows refer to the data flow of feature engineering.

**Figure 3 entropy-25-01549-f003:**
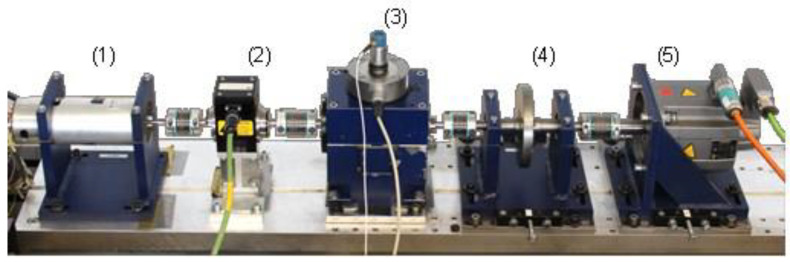
The experimental setup for rolling bearing fault diagnosis. (1) electric motor, (2) torque-measuring shaft, (3) bearing test module, (4) flywheel, (5) load motor.

**Figure 4 entropy-25-01549-f004:**
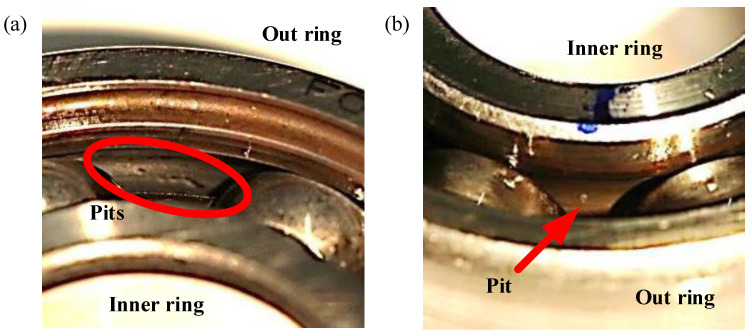
Real damages caused by accelerated lifetime tests: (**a**) Indentation at the raceway of outer ring; (**b**) small pitting at the raceway of inner ring.

**Figure 5 entropy-25-01549-f005:**
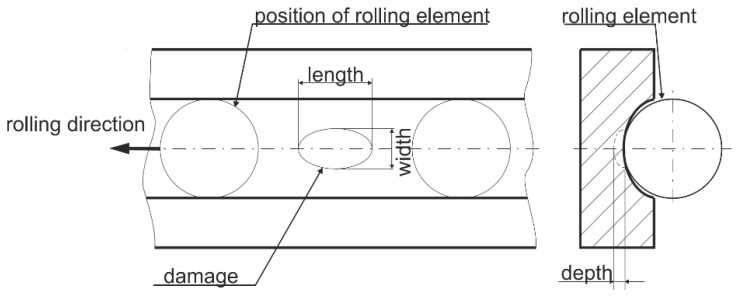
Parameters for describing the geometry of bearing damages.

**Figure 6 entropy-25-01549-f006:**
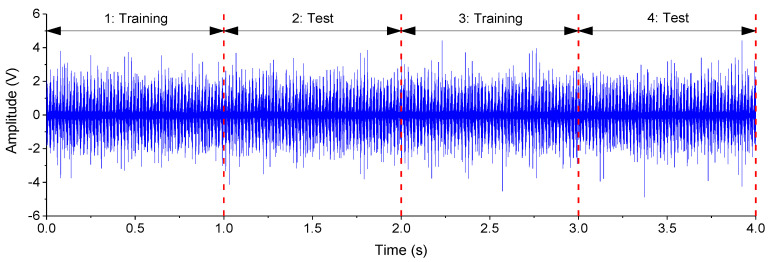
Vibrations of the bearing test module (N09_M07_F10_K001_2).

**Figure 7 entropy-25-01549-f007:**
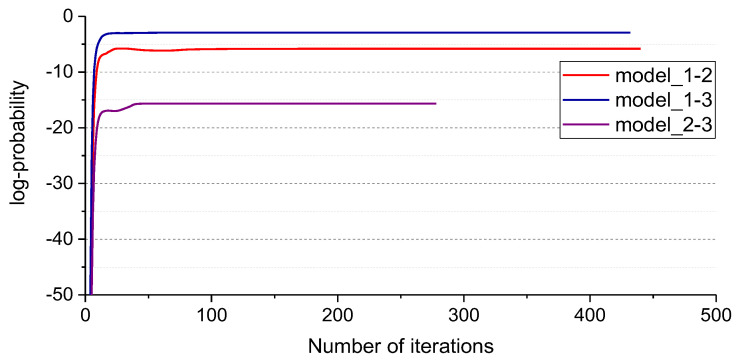
The iterative process of Standard_SBC by using the 126 TFW features obtained from the phase currents $\left(u,v\right)$ and vibrations.

**Figure 8 entropy-25-01549-f008:**
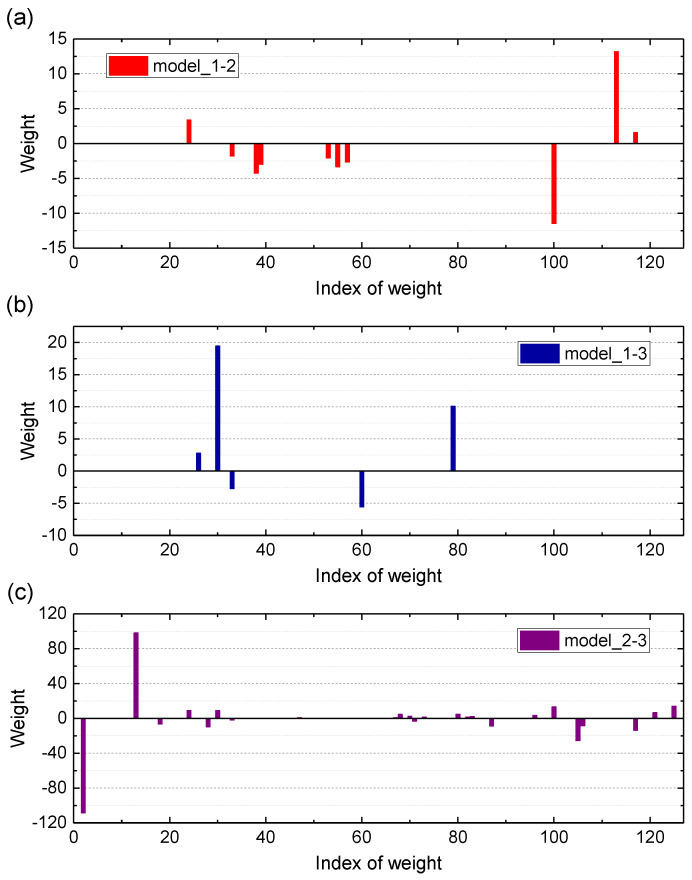
The weights $w_{i}$ of the SBC-based rolling bearing fault diagnosis system constructed by the 126 TFW features obtained from the phase currents $\left(u,v\right)$ and vibrations.

**Figure 9 entropy-25-01549-f009:**
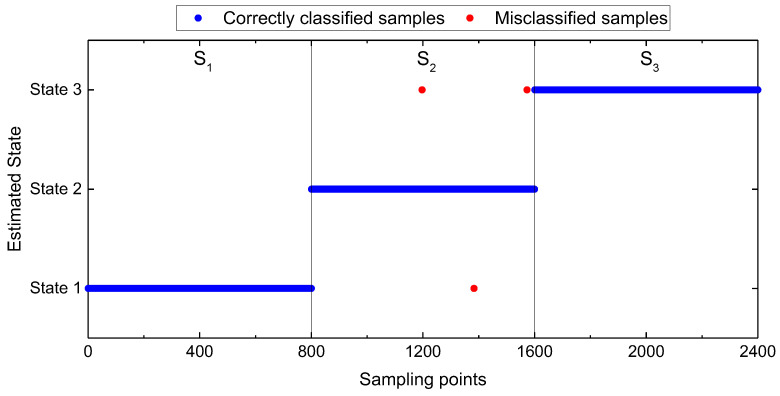
Test results of the constructed SBC-based rolling bearing fault diagnosis system by using the 126 TFW features obtained from the phase currents $\left(u,v\right)$ and vibrations.

**Figure 10 entropy-25-01549-f010:**
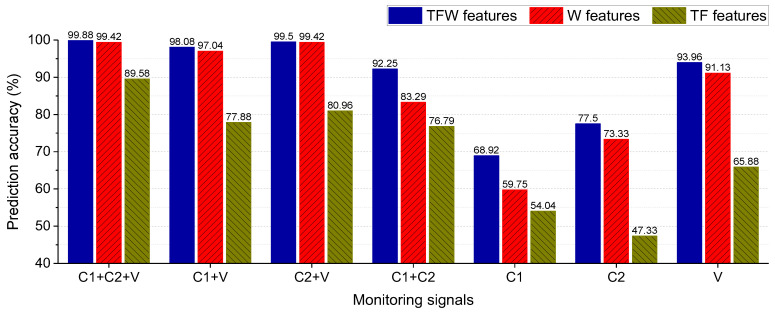
Performance evaluation of the SBC-based rolling bearing fault diagnosis system for different combinations of signals and features. ‘T’, ‘F’, and ‘W’ represent the time-domain, the frequency-domain, and the wavelet-domain, respectively. C1 and C2 represent the phase currents u and v of motors, respectively. V represents the vibration signal of bearings.

**Figure 11 entropy-25-01549-f011:**
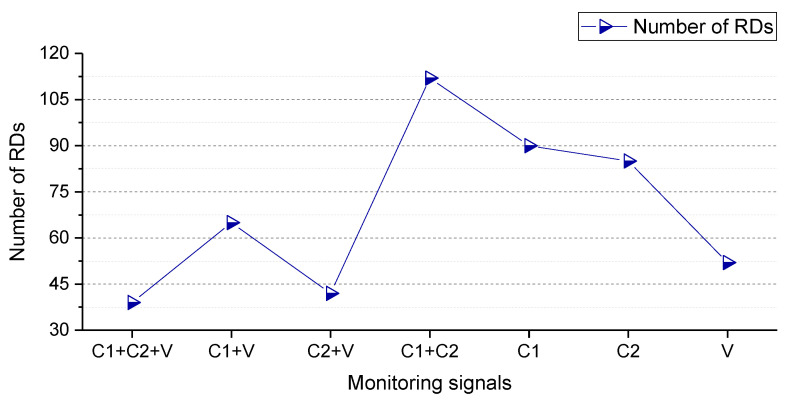
Sparseness of the SBC-based rolling bearing fault diagnosis system by utilizing the TFW features.

**Figure 12 entropy-25-01549-f012:**
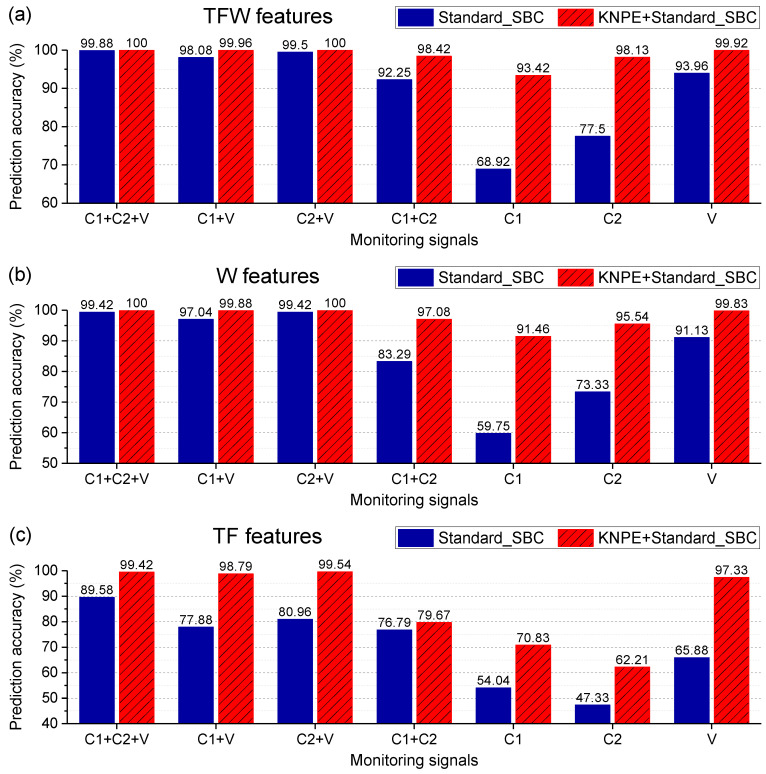
Effectiveness analysis of KNPE for rolling bearing fault diagnosis (k=10, $\sigma_{0}=5$, γ=50%): (**a**) KNPE fuses the TFW features; (**b**) KNPE fuses the W features; (**c**) KNPE fuses the TF features.

**Figure 13 entropy-25-01549-f013:**
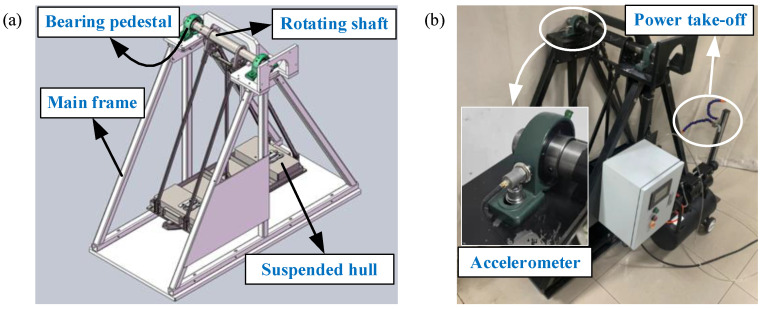
The experimental setup for rotating shaft fault diagnosis: (**a**) Three-dimensional model; (**b**) actual installation.

**Figure 14 entropy-25-01549-f014:**
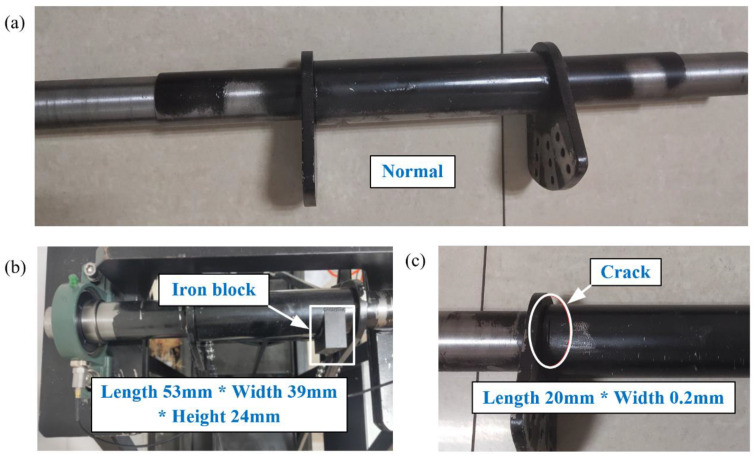
Faulty shafts caused by artificial damages: (**a**) Normal shaft; (**b**) unbalanced shaft; (**c**) cracked shaft.

**Figure 15 entropy-25-01549-f015:**
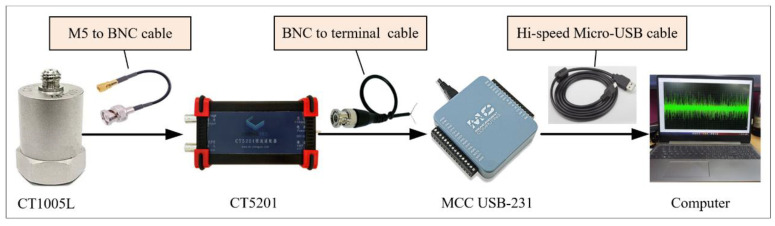
Connection of key components; BNC (Bayonet Nut Connector).

**Figure 16 entropy-25-01549-f016:**
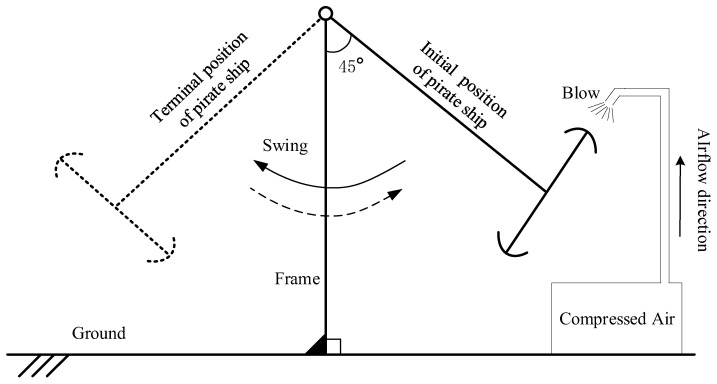
The Pirate Ship Data Collection Experiment.

**Figure 17 entropy-25-01549-f017:**
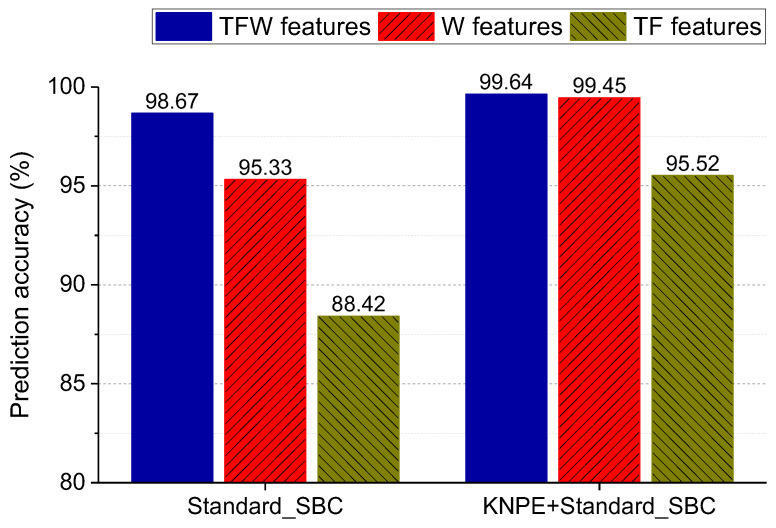
Effectiveness analysis of KNPE for rotating shaft fault diagnosis (k=10, $\sigma_{0}=5$, γ=2%). ‘T’, ‘F’, and ‘W’ refer to the time-domain, the frequency-domain, and the wavelet-domain, respectively.

**Figure 18 entropy-25-01549-f018:**
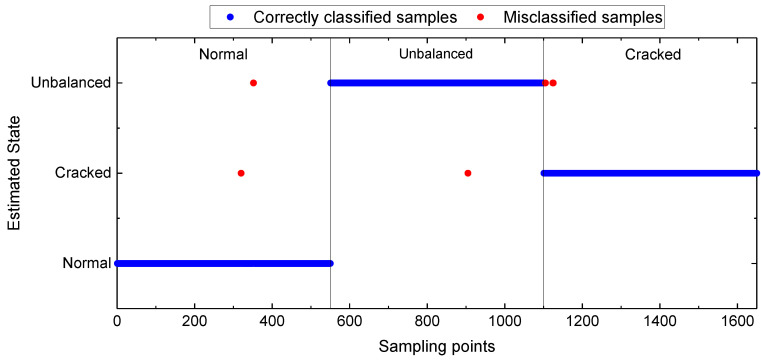
Test results of the rotating shaft fault diagnosis system based on KNPE and Standard_SBC by using the TFW features (k=10, $\sigma_{0}=5$, γ=2%).

**Figure 19 entropy-25-01549-f019:**
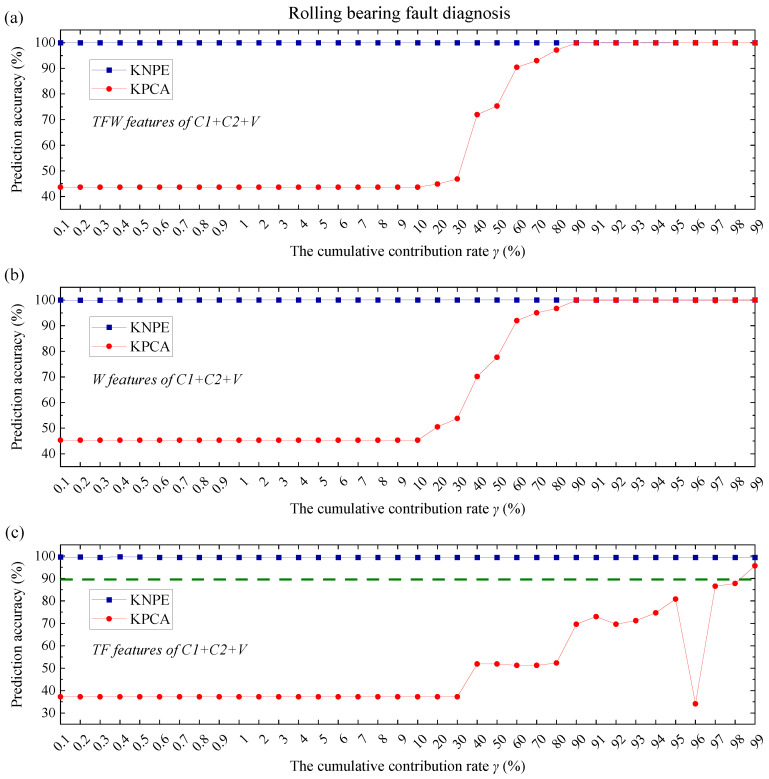
Performance comparison of KNPE (k=10, $\sigma_{0}=5$) and KPCA ($\sigma_{0}=5$) for rolling bearing fault diagnosis: (**a**) KNPE fuses the TFW features of C1+C2+V; (**b**) KNPE fuses the W features of C1+C2+V; (**c**) KNPE fuses the TF features of C1+C2+V.

**Figure 20 entropy-25-01549-f020:**
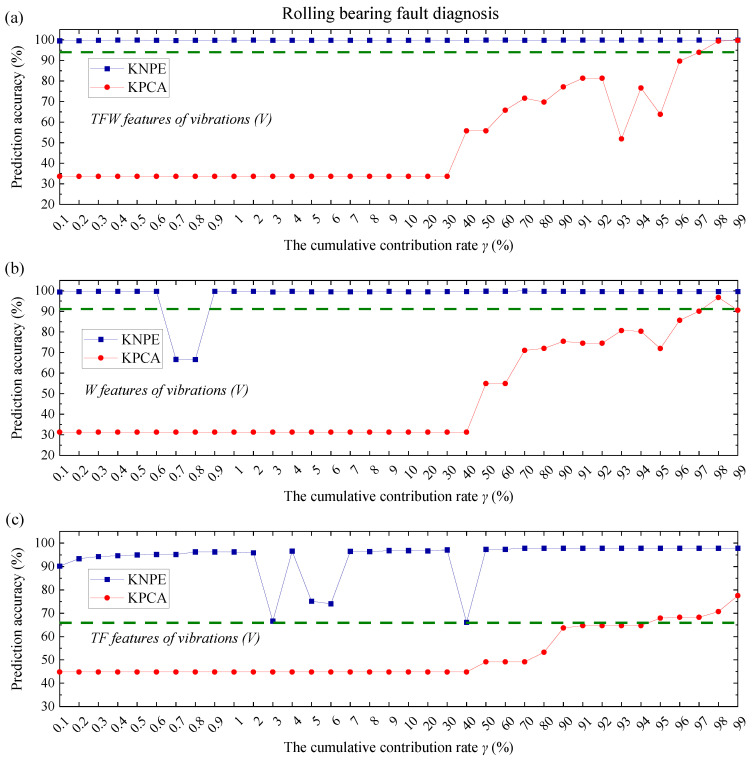
Performance comparison of KNPE (k=10, $\sigma_{0}=5$) and KPCA ($\sigma_{0}=5$) for rolling bearing fault diagnosis: (**a**) KNPE fuses the TFW features of vibrations (V); (**b**) KNPE fuses the W features of vibrations (V); (**c**) KNPE fuses the TF features of vibrations (V).

**Figure 21 entropy-25-01549-f021:**
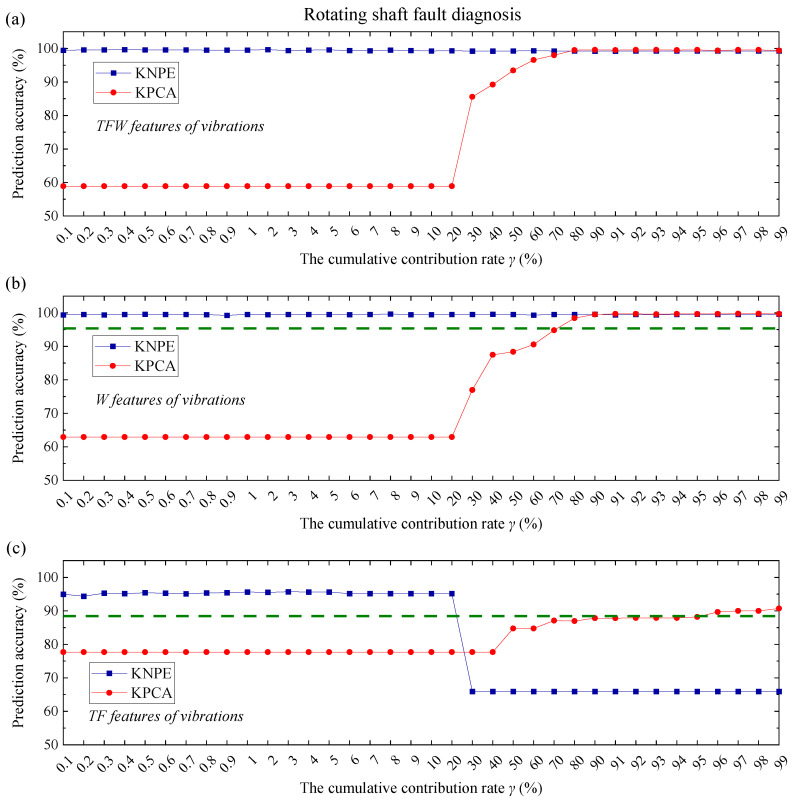
Performance comparison of KNPE (k=10, $\sigma_{0}=5$) and KPCA ($\sigma_{0}=5$) for rotating shaft fault diagnosis: (**a**) KNPE fuses the TFW features of vibrations; (**b**) KNPE fuses the W features of vibrations; (**c**) KNPE fuses the TF features of vibrations.

**Table 1 entropy-25-01549-t001:** Three domains and the corresponding features.

Domains	Features [21]	Expressions
Time-domain (T)	Mean	$\mu =E\left(\left|x_{i}\right|\right)$
	Root mean square (RMS)	$x_{RMS}={\left\{E\left(x_{i}^{2}\right)\right\}}^{1/2}$
	Maximum (Max)	$x_{Max}=max\left(\left|x_{i}\right|\right)$
	Peak to valley (PV)	$x_{PV}=max\left(\left|x_{i}\right|\right)-min\left(\left|x_{i}\right|\right)$
	Standard deviation (Std)	$x_{Std}=\sigma ={\left\{E\left[{\left(\left|x_{i}\right|-\mu \right)}^{2}\right]\right\}}^{1/2}$
	Skewness (Ske)	$x_{Ske}=E\left\{{\left[\left(\left|x_{i}\right|-\mu \right)/\sigma \right]}^{3}\right\}$
	Kurtosis (Kur)	$x_{Kur}=E\left\{{\left[\left(\left|x_{i}\right|-\mu \right)/\sigma \right]}^{4}\right\}$
	Form factor (Fmf)	$x_{Fmf}=x_{RMS}/\mu$
Frequency-domain (F)	Frequency centroid (FC)	$x_{FC}=\frac{\sum_{i=1}^{N}f_{i}\cdot P\left(f_{i}\right)}{\sum_{i=1}^{N}P\left(f_{i}\right)}$
	Frequency variance (FV)	$x_{FV}=\frac{\sum_{i=1}^{N}{\left(f_{i}-x_{FC}\right)}^{2}\cdot P\left(f_{i}\right)}{\sum_{i=1}^{N}P\left(f_{i}\right)}$
Wavelet-domain (W)	RMS of the wavelet packet coefficients [33]	${RMS}_{F_{dir}-L-M}=\sqrt{\frac{1}{N_{i}}\sum_{t=1}^{N_{i}}{\left(d_{j-L,t}^{\left(M\right)}\right)}^{2}}$

**Table 2 entropy-25-01549-t002:** Pseudo-code of the feature fusion algorithms (NPE and KNPE).

	NPE	KNPE
Calculation of the eigenvectors and eigenvalues		
1	Constructing the adjacency graph G	
2	Calculation of the weight matrix W as given by Equation (1)	
3	Calculation of the semi-positive definite matrix M as given by Equation (3)	
4	×	Constructing the kernel matrix $K_{center}$ as given by Equation (10)
5	×	Constructing the kernel matrix $K_{test}^{C}$ as given by Equation (14)
6	Calculation of the eigenvectors and eigenvalues by solving Equation (2)	Calculation of the eigenvectors and eigenvalues by solving Equation (9)
Calculation of the fused features		
1	Determination of the dimension d of the fused features	
2	Calculation of the fused features for the training data by Equation (4)	Calculation of the fused features for the training data by Equation (13)
3	Calculation of the fused features for the test data by Equation (5)	Calculation of the fused features for the test data by Equation (15)

**Note:** In step 6, the singular value decomposition (SVD) [34,35] is applied for improving the computing efficiency of the feature fusion. Detailed process is provided in Appendix A.

**Table 3 entropy-25-01549-t003:** Categorization for healthy bearings and bearings with real damages.

Actual State	Healthy (Class 1)	Outer Ring Damage (Class 2)	Inner Ring Damage (Class 3)
	K001	KA04	KI04
	K002	KA15	KI14
Bearing code	K003	KA16	KI16
	K004	KA22	KI18
	K005	KA30	KI21

**Table 4 entropy-25-01549-t004:** Operating parameters of the test rig.

No.	Rotational Speed (rpm)	Load Torque (Nm)	Radial Force (N)	Name of Setting
1	900	0.7	1000	N09_M07_F10
2	1500	0.1	1000	N15_M01_F10
3	1500	0.7	400	N15_M07_F04
4	1500	0.7	1000	N15_M07_F10

**Table 5 entropy-25-01549-t005:** Operating parameter of healthy (undamaged) bearings during run-in period.

No.	Bearing Code	Run-in Period (h)	Radial Force (N)	Speed (min^−1^)
1	K001	>50	1000–3000	1500–2000
2	K002	19	3000	2900
3	K003	1	3000	3000
4	K004	5	3000	3000
5	K005	10	3000	3000

**Table 6 entropy-25-01549-t006:** Test bearings with real damages caused by accelerated lifetime test.

No.	Bearing Code	Type of Damage	Damage Location	Combination	Arrangement	Damage Level	Characteristic of Damage
1	KA04	fatigue: pitting	OR	S	no repetition	1	single point
2	KA15	plastic deform.: Indentations	OR	S	no repetition	1	single point
3	KA16	fatigue: pitting	OR	R	random	2	single point
4	KA22	fatigue: pitting	OR	S	no repetition	1	single point
5	KA30	plastic deform.: Indentations	OR	R	random	1	distributed
6	KI04	fatigue: pitting	IR	M	no repetition	1	single point
7	KI14	fatigue: pitting	IR	M	no repetition	1	single point
8	KI16	fatigue: pitting	IR	S	no repetition	3	single point
9	KI18	fatigue: pitting	IR	S	no repetition	2	single point
10	KI21	fatigue: pitting	IR	S	no repetition	1	single point

**Note:** OR: outer ring; IR: inner ring; S: single damage; R: repetitive damage; M: multiple damage. Single damage: One single component of the rolling bearing is affected by a single damage. Repetitive damage: Identical damage symptoms are repeated at several places on the same bearing component. Multiple damage: Different damage symptoms occur in the bearing or identical damage symptoms occur on different bearing components.

**Table 7 entropy-25-01549-t007:** Damage levels determine the extent of damage.

Damage Level	Assigned Percentage Values	Limits for Bearing 6203
1	0–2%	≤2 mm
2	2–5%	>2 mm
3	5–15%	>4.5 mm
4	15–35%	>13.5 mm
5	>35%	>31.5 mm

**Table 8 entropy-25-01549-t008:** The feature extraction of phase currents (u,v) and vibrations.

No.	Domain	Symbol	Feature Number
1	Time-domain	T	24
2	Frequency-domain	F	6
3	Wavelet-domain	W	96
4	Total	TFW	126

**Table 9 entropy-25-01549-t009:** The hyper-parameters of Standard_SBC-based fault diagnosis system.

No.	Hyper-Parameters	Dimension	Initial Value
**1**	w	*M*	$w_{0}=w_{1}=\cdots =w_{M}=0$
2	α	*M*	$\alpha_{0}=\alpha_{1}=\cdots =\alpha_{M}=5\times 10^{-5}$
3	β	*N*	$\beta_{1}=\beta_{2}=\cdots =\beta_{N}=0.25$

**Table 10 entropy-25-01549-t010:** The number of features extracted by utilizing the different signals and domains.

Signal	Domain		
	TF	W	TFW
C1	30	96	126
C2	30	96	126
V	30	96	126
C1 + C2	60	192	252
C1 + V	60	192	252
C2 + V	60	192	252
C1 + C2 + V	90	288	378

**Table 11 entropy-25-01549-t011:** The type and function of the key components of the experimental platform.

No.	Name	Type	Parameter	Function
1	Accelerometer	CT1005L	Measure range: ±100 g, Frequency response: 1~10 kHz	Convert vibration signals into voltage signals.
2	Constant current source	CT5201	Frequency range: 1~100 kHz	Stabilize the voltage signals.
3	Data acquisition card	MCC USB-231	Analog input range: ±10 V, Resolution ratio: 16-bit; Sample frequency: 50 kS/s	Collection of the voltage signals.

**Table 12 entropy-25-01549-t012:** Categorization for the three shafts and the corresponding amount of data that was collected by the accelerometer.

	Normal Shaft	Unbalanced Shaft	Cracked Shaft
Categorization	Class 1	Class 2	Class 3
Amount of data	110	110	110

**Table 13 entropy-25-01549-t013:** Comparison between Standard_SBC and RVM.

Case Studies	Methods	Accuracy (%)	CV_Time (s)	Training_Time (s)	Test_Time (s)
Rolling bearing fault diagnosis	Standard_SBC	99.88	×	13.5940	0.0090
	RVM	99.96	$1.0669\times 10^{4}$	42.3480	0.1560
	KNPE + Standard_SBC	100	×	0.2700	0.0090
	KNPE + RVM	100	$2.7469\times 10^{4}$	48.9790	0.2050
Rotating shaft fault diagnosis	Standard_SBC	98.67	×	4.3220	0.0040
	RVM	99.52	$4.8212\times 10^{3}$	28.8570	0.0220
	KNPE + Standard_SBC	99.58	×	17.3770	0.0050
	KNPE + RVM	99.16	$1.1088\times 10^{4}$	13.6910	0.1000

**Note:** As for rolling bearing fault diagnosis, TFW features of C1+C2+V are adopted. As for rotating shaft fault diagnosis, TFW features of vibrations are adopted. As for KNPE (k=10, $\sigma_{0}=5$), the parameter γ is set to 0.2%.

## Data Availability

Data are contained within the article.

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
