# Peer review of "Fault Diagnosis of Rotating Machinery Using Kernel Neighborhood Preserving Embedding and a Modified Sparse Bayesian Classification Model"

_entropy, 2023, doi:10.3390/e25111549_

Round 1
Reviewer 1 Report
Comments and Suggestions for Authors
Please find the attached file.

The manuscript would benefit from simplifying its grammar usage, which can improve the article's readability. Consider refining the language to be clearer, more nuanced, and smoother for the reader.
Reviewer 2 Report
Comments and Suggestions for Authors+X_C/V

Reviewer 3 Report
Comments and Suggestions for Authors
Comment to the Authors
1.-Bearing tests number
How many tests have you used in your research work?
It is not clear the data from references 28 and 29 are used or not? Please could you clarify this issue?
“…
In this section, the experimental data is obtained from the Bearing Data Center of Paderborn University [28, 29].
…”
“…
Real damages caused by accelerated lifetime tests as shown in Fig. 4 are comprehensively investigated in this work.
…”
As far as I understand you have modified 15 bearing, and you have performed 20 measurements on each one.
2.- Fault severity of the performed tests in bearings
Which are the fault severity level of the performed tests in the bearing?
Have your experimental test in healthy conditions?
3.- Shaft tests experimental setup
Does the experimental setup produce similar conditions to a real machine?
4.- Fault severity of the performed tests in shafts
You have used three different shafts, healthy, unbalance and cracks.
Which are the fault severity level of the performed tests in the shafts for unbalance and cracks?
Reviewer 4 Report
Comments and Suggestions for Authors
In general, the scientific sound of the article is weak. The method is not presented clearly enough to get the main points about the research. The motivation of the article is exaggerated regarding human losses because of not detected bearing faults in rotary machinery.
In section 3.1, first paragraph, an explanation for the normalization is missing. What is the scientific reason to normalize the data? Also, the features named in reference [21] shall be listed in a table in the appendix to get an overview.
Figure 2 doesn't clearify the algorithm and should be revised. The process is not clear.
In section 4.1.2, first and second paragraph, an amount of different features is mentioned. What's the reason for this amount of features? Why are they chosen? What are the features exactly? A summery in a table would be helpful.
The first paragraph of section 4.1.3.1 is an unnecessary repetition. The fifth paragraph is not specific enough and the figures 6-8 are not described in a sufficient way. At the end of the section, what are the explanations so that the conclusions can be drawn? Anything could be found in that way. The first two conclusions are not new and generally known.
There is the general question, if the results are compared to other common methods or is the comparison only internal?
In section 4.2.2, first paragraph, an amount of different features is mentioned. What's the reason for this amount of features? Why are they chosen? What are the features exactly? A summery in a table would be helpful.
In general, the additional value of this work is not clear. What is the aim? Why is this an important approach? What is better in an international comparison? This is totally missing and should be clearified, otherwise it is not possible to accept the article.
Comments on the Quality of English LanguageThe grammar has to be improved as well as the general English of the paper.
Round 2
Reviewer 1 Report
Comments and Suggestions for Authors
Perfectly revised, thanks for the effort.
Author Response
Dear reviewer,
Thank you very much for your affirmation of our paper, and grateful for your efforts. The improvement of our manuscript benefits from your valuable feedback. Once again, we grateful for your affirmation and feedback, sincerely.
Sincerely!
Reviewer 3 Report
Comments and Suggestions for Authors
Comment to the Authors
Thank you very much for your replay.
1.-Bearing tests number
Thank you very much for your answer.
2.- Fault severity of the performed tests in bearings
Thank you very much for your answer.
3.- Shaft tests experimental setup
This issue is still not clear for me. Please let me ask some questions to clarify this issue:
Does in the experimental setup the shaft rotate? If yes, have you tried at different speeds?
What is the torque applied in the test?
Which are the frequencies and amplitudes of the acceleration measurements?
What is the purpose of the power take-off?
4.- Fault severity of the performed tests in shafts
You have used three different shafts, healthy, unbalance and cracks.
Which are the fault severity level of the performed tests in the shafts for unbalance and cracks?
According to your replay there only one crack and one unbalance.
How many tests have you performed with the shafts? (110 tests each, but at the same fault severity level)
In my opinion you should better describe the experimental setup and the test procedure.
It should be useful to understand the paper to include some records of the vibrations.
Reviewer 4 Report
Comments and Suggestions for Authors
All comments have been adressed and the paper has been revised successfully
Comments on the Quality of English LanguageLittle proveread necessary.
Author Response
Dear reviewer,
Thanks for your affirmation and efforts of our paper. Benefit from your sincerely comments, the quality of our manuscript has been improvement significantly. This time, we have further polished the writing and grammar of our manuscript and the changes marked in green. Once again, we would like to express our gratitude for your efforts of our paper.
Sincerely!
Round 3
Reviewer 3 Report
Comments and Suggestions for Authors
Comment to the Authors
Thank you very much for your replay.